# Structural Stability Monitoring of Model Test on Highway Tunnel with Lining Backside Voids Using Dynamic and Static Strain Testing Sensors

**DOI:** 10.3390/s23031403

**Published:** 2023-01-26

**Authors:** Chaofei Du, Chuanbo Zhou, Nan Jiang, Yiwen Huang

**Affiliations:** Faculty of Engineering, China University of Geosciences, Wuhan 430074, China

**Keywords:** highway tunnel, lining void, model test, structural stability

## Abstract

Voids behind a lining may develop due to insufficient backfilling, poor workmanship, water erosion or gravity. They affect the interaction between the surrounding rock and lining and even cause instability of the lining structure. To ensure the safe operation of tunnels, it is very important to study the influence of voids behind the lining of the lining structure. In this paper, a laboratory model of a tunnel lining was established by taking the voids behind the lining of the Wushan Tunnel as an example. By changing the position and size of the voids, the corresponding stress variation law of the lining was obtained, and the influence of the voids behind the lining on the structural stability of the highway tunnel was analyzed. The experimental results showed that the voids behind the lining led to an increase in the stress near the voids, especially the voids at the vault. The circumferential stress and axial stress increased with increasing void depth and length, and the increase was greater with increasing void depth than increasing length; that is, the void depth had a greater effect on the lining stress. When the vault void depth was 30 mm, the axial tensile stress of the vault was 0.281 MPa, and the maximum increase was 178.2% compared with that without voids. The safety factors at different lining positions, from large to small, are: arch foot > spinner > arch top > arch waist. In the processes of lining operation and maintenance, special attention should be given to the treatment of voids behind the lining, especially deep voids.

## 1. Introduction

In recent years, as the state strengthens its investment in infrastructure construction, China’s transportation construction has undergone rapid development. At present, China is the country with the largest number of tunnel projects, the most complex structure and the fastest development speed in the world [1]. However, with the use of highway tunnels, various diseases often occur in the older tunnels, among which the void behind the tunnel lining is one of the most common tunnel diseases [2]. Zhang et al. [3] investigated about 100 railway tunnels in China and found that nearly 11.56% of tunnels had contact loosening and cavities behind the lining. The existence of lining voids reduces the stability of the lining structure, threatens the safety of driving in the tunnel and shortens the maintenance cycle and service life of a highway tunnel. Therefore, it is important to analyze the influence of the voids behind the lining on the structural stability of the operational highway tunnels and evaluate their safety.

A lot of research on the void disease behind tunnel linings has been conducted by scholars at home and abroad. Some scholars have conducted theoretical analysis on the stratum voids problem and obtained the calculation formula of surrounding rock stress and lining internal force when there is a void behind the lining [4,5,6]. In addition, some scholars have used finite element analysis software, such as ABAQUS, ANSYS and MIDAS-GTS, to study the void disease behind the lining [7,8,9,10,11,12]. Zhang et al. [13] and Bao et al. [14] established a three-dimensional numerical model to study the influence of the geometric size of the tunnel void on the internal force and safety of the lining structure. Ye [15] used numerical simulation to study the influence of the voids and loose contact state between support and surrounding rock on the safety of the lining structure. Li et al. used ANSYS software to study the mechanical behavior of the tunnel structure when there are cavities of different shapes and sizes behind the vault lining [16,17,18]. Min et al. [19] studied the mechanical characteristics of a double-arch tunnel under the action of a void at the top of the middle wall through numerical simulation.

Due to the complexity of tunnel engineering and the operating environment [20], the interaction between the surrounding rock and the lining is not clear at present. If only theoretical analysis and numerical simulation and other technical means are used for research, there are bound to be shortcomings because the limitations of the numerical simulation itself. For example the theoretical framework and boundary condition hypothesis cannot fit the reality. Therefore, some scholars used indoor model tests to conduct further research on the voids behind the lining [21,22,23,24,25]. Zhang et al. [26] studied the evolution law of tunnel structure cracks under the condition with double cavities in the tunnel vault and the back of the arch through a 1:70 indoor model test. In recent years, some scholars have studied the influence of void defects behind shield tunnel composite lining on structural mechanical characteristics and contact pressure with the stratum [27,28,29,30,31]. Leung [32] simulated the initial pressure of a shield tunnel with a test device separating the lining from the surrounding soil at different positions to simulate a void and studied the influence of the void behind the lining on the earth pressure distribution on the tunnel lining. Some scholars even used theoretical analysis, numerical simulation and model tests to carry out related research [2,33,34,35,36]. Zhang et al. [37] used numerical simulations and model tests to study the safety state of the tunnel structure under the condition of double cavities behind the vault and the arch shoulder, respectively. Some scholars have improved tunnel disease detection methods for tunnel disease detection. For example, a multi-layer SAFT high precision ultrasonic imaging method was proposed for void disease detection [38]. Yue et al. [39] put forward a method to calculate shield tunnel displacements of a full cross-section tunnel.

Many scholars have conducted studies of the stability of tunnels with defective engineering. Much of their research has investigated the influence of voids behind the lining on the safety of the lining structure. However, most of those studies have focused on numerical simulations and shield tunnel research. Too many assumptions have been made in those studies, which made it difficult to accurately describe the development of tunnel defects under actual working conditions. More studies are needed to verify the universality and rationality of the results of those studies. It is difficult to prevent the influence of size effects due to the small size of most laboratory tests. Therefore, a horizontal loading test device was designed to simulate the surrounding rock pressure through jacks in light of the complex original working conditions of the tunnel. To avoid the influence of size effects, a large scale model (1:10) is selected in this paper to simulate a typical mountain highway tunnel damage project. The influence of the voids behind the lining on the stability of the lining structure are systematically simulated for different sizes and positions of the cavity and analyzed experimentally, providing references for the maintenance and reinforcement of tunnels with void disease.

## 2. Similar Model Test

### 2.1. General Information of the Tunnel Project

The Wushan Tunnel, located at the southeast part of Gansu Province, is an important part of the Tianshui to Dingxi section of the National Highway G30, as shown in Figure 1. The upper line of the tunnel is 2.5 km long, with a maximum depth of 270 m. The tunnel mainly passes through the surrounding rock of grade Ⅳ and Ⅴ, and the engineering geological conditions are quite complex. As shown in Figure 2, the secondary lining prototype section of the tunnel is a two-lane four-center circle. The section size of the lining is 1186 cm wide, 963 cm high, 50 cm thick concrete equal section structure. The lining section is symmetrical left and right, so only the radius and radian of arcs on the right side of the section are marked in the figure. According to the inspection, there are 57 voids behind the secondary lining of the upper tunnel, with a cumulative length of 244.0 m, accounting for 9.8% of the total length.

### 2.2. Similarity-Scaling Relationship

In this paper, the geometric similarity ratio of the prototype and model was set as C_L_ = 10. Using this as the basic similarity ratio, we derive the similarity ratios of the prototype and model for each of the physical and mechanical parameters according to the similarity theory: heavy similarity ratio C_γ_ = 1, Poisson’s ratio C_μ_ = 1, internal friction angle C_ϕ_ = 1, elastic modulus similarity ratio C_E_ = 10 and cohesive force similarity ratio C_c_ = 10.

### 2.3. Similar Materials and Similar Models

In the safety model tests of the lining structure, gypsum was used as a material similar to plain concrete. Gypsum, as a common brittle material, is similar to concrete in fracture mechanics, so it is an ideal elastic model material. The secondary lining of the tunnel prototype is a C25 concrete structure, and the mechanical parameters were set according to the actual engineering values. The elastic modulus was 28 GPa, the ultimate compressive strength was 16.7 MPa, and the ultimate tensile strength was 1.78 MPa. In this paper, a mixture of gypsum and water was used to simulate the lining structure. Lining of according to the model To obtain the physical and mechanical parameters for the direct shear test and the compression test for the model, we adjusted the lining material proportioning of water and gypsum according to a similarity ratio of 0.6:1 mixture. Model physical and mechanical parameters are shown in Table 1. In addition to the specimen density and C25 concrete, ideal similar material bulk density was large, but severe differences have little impact on the content of the study for the test, Other physical and mechanical parameters of the model material basically meet the test needs. The material ratio of the tunnel secondary lining structure similarity model was water: gypsum = 0.6:1, and the specific parameters are shown in Table 1.

As shown in Figure 3, the lining was prefabricated in a mold and maintained under certain temperature and humidity conditions after demolding. All reduced model section sizes were 1/10 of the prototype according to Figure 2; that is, the model geometry size was 1/10 of the prototype. The lining section and the thickness of the lining model was set as 0.05 m, the span was 1.19 m, the height was 0.96 m, and the axial length was 0.45 m. 

### 2.4. Tunnel Similarity Model

The self-made indoor model platform consists of the similar lining mode, a loading system and a data monitoring system. The tunnel lining–soil complex was used to simulate the actual working conditions, and the horizontal loading mode was adopted. The test system could simulate the dead weight stress field of the tunnel lining, and the surrounding rock was filled with clay to simulate the lining under uniform stress. The outer layer of the soil layer was enclosed by a 1 cm thick steel plate, a pressurizing system composed of a jack, and the counter-force frame was welded with I-beam steel to simulate the surrounding rock to provide reaction force. The entire model diagram is shown in Figure 4. The actual picture of the model is shown in Figure 5.

#### 2.4.1. Loading System

The loading system consists of a pressure jack and a reaction steel frame, as shown in Figure 4. The test jacks were FCY-10100 hydraulic jacks, and the specifications were 10T horizontal loading hydraulic jacks. Each jack was equipped with a CP-180 manual pump, 1.5-m oil pipe and a pressure gauge. The vertical uniform pressure on the lining structure was simulated by jacks J3 and J4 located at the vault of the tunnel lining model. The horizontal distribution pressure on the tunnel lining structure was simulated by J1 and J2 on the left side of the tunnel lining and J5 and J6 on the right side of the tunnel lining. The jacks converted the point load into a uniform load to act on the lining structure of the tunnel through the 1cm thick steel plate at the front end and the silty clay medium between the steel plate and the lining structure. 

The earth pressure sensors P1, P2, P3 and P4 were, respectively, fixed at the left arch waist, the left side of the arch, the right side of the arch and the right arch waist on the surface of the lining structure and connected with the DH5956 dynamic signal test and analysis system. The accurate loading of the tunnel lining structure was achieved based on the readings of the earth pressure sensor. The lower parts of the jacks were in contact with the reaction frame, and the model was loaded by the reaction force provided by the frame.

#### 2.4.2. Data Monitoring System

The data monitoring system consists of several strain gauges, earth pressure gauges, two DH3817 dynamic and static strain testing systems, and one DH5956 strain collection analyzer. This test mainly monitors the stress change of the tunnel lining under the action of ground stress and uses a 120-50AA resistive strain gauge for measurement. As shown in Figure 4, strain gauges S1–S6 were pasted on the left wall, left arched waist, left arched shoulder, vault, right arched shoulder, right arched waist and right wall of the inside the lining structure. A group of strain flowers will be added to the void when the void behind the lining is tested. 

Figure 6 shows the working principle of the DH3817. The system can realize sampling, transmission, storage, and display at the same time and can use a mass computer storage hard disk to record multi-channel signals for a long time without interruption. In this paper, a DH3817 dynamic and static stress and strain measurement (Taizhou City, Jiangsu Province, China) and analysis system is used to collect the strain gauge data.

### 2.5. Pressure of Model Test

The Wushan Tunnel is mainly buried deep and is part of a long tunnel buried in the mountains, and there is no bias pressure and expansion force in the surrounding rock. This test only considers the simulation of ground stress conditions of the tunnel under the deep buried condition. According to the Highway Tunnel Design Code Volume I Civil Engineering (JTG 3370.1-2018) [40], the vertical and horizontal uniform pressure of the loose load in a deep tunnel can be calculated according to the following formula under the condition of surrounding rock without significant bias and expansion force:

#### 2.5.1. The Vertically Distributed Pressure 

The vertically distributed pressure can be calculated according to the following equation
(1)q=γh
(2)$$h=0.45\times 2^{s-1}\omega$$
where *q* is the vertical uniform pressure; kN/m^2^; *γ* is the surrounding rock weight; kN/m^3^; *h* is the height of the surrounding rock pressure calculation, m; *s* is the level of the surrounding rock, with integer values of 1, 2, 3, 4, 5 and 6; *ω* is the width influence coefficient, calculated as follows: *ω* = 1 + *i* (*B* − 5); *B* is the tunnel width, and m; *i* is the increase or decrease rate of the surrounding rock pressure when the tunnel width increases or decreases by 1m, as shown in Table 2 and as can be seen from the design data of the upper line of the Wushan Tunnel, *i* = 0.12.

#### 2.5.2. Horizontal Surrounding Rock Pressure

The horizontal surrounding rock pressure of the deep tunnel can be valued according to Table 3, and the horizontal distribution pressure in this test is set as e = 0.5*q*.

According to the geological conditions of the Wushan Tunnel, the surrounding rock weight is 20 kN/m^3^. The vertical distribution pressure of the surrounding rock is *q* = 0.263 MPa, and the horizontal distribution pressure e = 0.5*q* = 0.132 MPa can be obtained.

### 2.6. Test Loading Scheme

This model test was mainly based on the disease of the Wushan Tunnel. According to the possible location and size of the hole behind the tunnel, a step-by-step loading method was adopted to simulate the following parameters: simulation of voids at different positions behind the lining, different depths behind the vault and different lengths behind the vault. Nine test conditions were set up to study the stress characteristics of the void defect lining, as shown in Table 4.

#### 2.6.1. Voids of Different Depths behind the Vault

In working conditions 2, 3 and 4, a void with a depth of 1 cm, a void with a depth of 2 cm and a void with a depth of 3 cm were successively installed on the outer wall of the vault, as shown in Figure 7a–c. In addition, seven groups of strain gauges were affixed to the left wall, left arch waist, left arch shoulder, vault, right arch shoulder, right arch waist and right wall in the inner side of the lining structure according to the scheme, and one group of strain gauges was affixed to the void of the outer wall. The pressure system was controlled to slowly pressure the lining until the vault pressures P2 and P3 both reached the vertical distribution pressure *q* of 0.263 MPa, and the arch waist pressure P1 and P4 on both sides reached the horizontal distribution pressure e of 0.132 MPa, and the pressure was stopped.

#### 2.6.2. Voids of Different Lengths behind the Vault

Figure 8a–c correspond to working conditions 2, 5 and 6, respectively. Voids with a length of 5 cm, a length of 10 cm and a length of 15 cm were set successively on the outer wall of the vault. According to the scheme, seven groups of strain gauges were arranged on the inner wall of the void, and one group of strain gauges was arranged at the outer wall void. The pressure system was controlled to slowly pressure the lining until the arch pressures P2 and P3 reached the vertical distribution pressure q of 0.263 MPa, and the arch waist pressures P1 and P4 on both sides reached the horizontal distribution pressure e of 0.132 MPa, and the pressure was stopped.

## 3. Analysis of Test Results 

By controlling the loading system, the lining was slowly pressurized until T2 and T3 reached the vertically distributed pressure q = 0.263 MPa, and T1 and T4 reached the horizontally distributed pressure e = 0.132 MPa. The stress–strain data of the lining under different working conditions were obtained. According to the stress-–strain data and morphological changes of the tunnel lining under nine groups of test conditions, the influence process and corresponding law of void on the stability of lining structure under different working conditions were analyzed.

### 3.1. Stress Analysis of Tunnel Lining without Void

When there was no void behind the lining, the external surface of the lining was under pressure, and the internal surface was under positive tension (the same below). Figure 9 shows that under the condition of no void defect, the stress values of the lining vault and the arch shoulder are positive tensile stress, while the stress values of the arch waist and the arch foot are negative compressive stress. However, whether it was tensile stress or compressive stress, the circumferential stress was generally greater than the axial stress at the same monitoring point. The inner circumferential tensile stress of the vault was 0.506 MPa, and the inner axial tensile stress was 0.101 MPa. The peak values of circumferential and axial tensile stresses appeared at the vault, while the peak values of circumferential and axial compressive stresses appeared at the arch waist, which can explain why the inner wall of the vault was mainly damaged by tensile stress, and the inner wall of the arch waist was mainly damaged by extrusion.

### 3.2. Stress Analysis of Cavities at Different Positions

The size of the void was controlled to be 50 mm × 20 mm × 10 mm (length × width × depth), and the positions of the void behind the lining were transformed into vault, arch shoulder, arch waist and arch foot in order to obtain the circumferential and axial stress values at different positions of the lining, as shown in Figure 10 and Figure 11. (1) When the void was located at the vault, the circumferential tensile stress of the vault was 0.604 MPa, which increased by 19.37% compared with the condition without the vault void. The axial stress of the inner wall was 0.142 MPa, and the increase was 40.59% compared with that of the vault without void. When the void was located at the arch shoulder, the arch waist and the arch foot, compared with the lining without the void, the circumferential and axial stress values of the lining did not change much at the same monitoring position. It shows that the void has the greatest influence on the stress of the lining structure when it is located at the vault but has little influence on the stress of the lining structure when it is located at other positions. (2) When the voids were located at the vault, shoulder, waist and foot, the circumferential stresses at the void were −0.613 MPa, −0.543 MPa, −0.545 MPa and −0.468 MPa; the axial stresses were −0.0564 MPa, −0.0321 MPa, −0.0463 MPa and −0.0128 MPa. According to the test data, with the change of the location of the void, the stress value of voids in the descending order was: arch > arch waist > arch shoulder > arch foot. The presence of void disease will lead to different degrees of increase in axial force and bending moment near the void position.

### 3.3. Stress Analysis of Vaults with Different Void Depth

During the test, the length × width of the void behind the lining was kept unchanged at 50 mm × 20 mm, and the depths were changed to 10 mm, 20 mm and 30 mm, respectively. As shown in Figure 12 and Figure 13: (1) When the depth of the vault void was 20 mm, the circumferential tensile stress of the inner wall of the vault was 0.715 MPa, which was 41.3% higher than that without the void; the axial tensile stress of the inner wall of the vault was 0.165 MPa, an increase of 63.4% compared with that without void. (2) When the depth of the vault void was 30 mm, the circumferential tensile stress of the vault was 0.802 MPa, an increase of 58.5% compared with that without void; the axial tensile stress of the vault was 0.281 MPa, an increase of 178.2% compared with that without the void. Both the circumferential stress and the axial stress of the inner wall of the lining vault increase with the increase of the void depth, and the increase of the axial stress becomes larger. (3) As the depth increases to 10 mm, 20 mm and 30 mm, the circumferential stresses at the voids of the outer wall of the corresponding lining vault were −0.613 MPa, −0.787 MPa and −0.862 MPa, respectively; the axial stresses at the voids were −0.0564 MPa, −0.0712 MPa and −0.13 MPa, respectively; that is, both the circumferential stress and the axial stress at the void of the outer wall of the lining vault increase with the increase of the void depth. In summary, the increase in the void depth of the vault has a more obvious impact on the stress of the inner and outer walls of the lining vault.

### 3.4. Stress Analysis of Vault with Different Void Length 

During the test, the width × depth of the void behind the lining were kept unchanged from 20 mm × 10 mm, and the length of the void was changed to 50 mm, 100 mm and 150 mm, respectively. As shown in Figure 14 and Figure 15: (1) When the vault void length was 100 mm, the circumferential tensile stress of the vault was 0.655 MPa, which increased by 29.4% compared with that without void. The axial tensile stress of the vault was 0.152 MPa, which increased by 50.5% compared with that without the void. (2) When the vault void length was 150 mm, the circumferential tensile stress of the vault was 0.708 MPa, which was 39.9% higher than when there was no void; The axial tensile stress of the vault was 0.182 MPa, an increase of 80.2% compared to the absence of voids. Both the circumferential and axial stresses on the inner walls of the lining vault increased with the increase in the length of the void, and the increase of the axial stress was larger. (3) As the length increased sequentially to 50 mm, 100 mm and 150 mm, the circumferential stresses at the voids behind the corresponding lining vault were −0.613 MPa, −0.63 MPa and −0.662 MPa, respectively; and the axial stresses at the voids were −0.0564 MPa, −0.047 MPa and −0.0508 MPa, respectively. The circumferential stresses and axial stresses of the voids behind the lining vault had little change with the increase of the cavity length; that is, the stresses of the voids behind the lining vault had little influence on the change of the cavity length. In addition, the stresses in other locations of the lining were not greatly affected by changes in the length of the voids.

### 3.5. Variation Law of Axial Force and Bending Moment of Lining

According to the inner stress value σ_1_ and the outer stress value σ_2_ of the lining section, the unit section bending moment M and axial force N can be calculated as [41]:(3)$$M=bh^{2}\left(\sigma_{1}-\sigma_{2}\right)/12\;\;$$
(4)$$N=bh\left(\sigma_{1}+\sigma_{2}\right)/2$$
where *σ_1_* is the stress on the inner wall of the lining; σ_2_ is the stress of lining outer wall; *b* is the unit length, taken as 1000 mm; and *h* is the lining thickness, which is taken as 50 mm.

China’s Code for Design of Highway Tunnels (JTG 3370.1-2018) and Code for Design of Railway Tunnels (TB 10003-2005) both provide clear calculation formulas and measurement standards for the safety factor of tunnel linings. When the calculation result *K* ≥ 2, the steel bar does not reaches ultimate strength or concrete does not reaches ultimate compressive or shear strength, the structure is relatively safe. In contrast, a calculation result *K* < 2 means that the steel bar has reached the ultimate strength or the concrete has reached ultimate compressive or shear strength, the safety and stability of the steel bar and concrete are insufficient, and the secondary lining structure needs to be further strengthened.

The following equation provides the concrete safety state discrimination standard:(5)$$KN\leq \varphi \alpha R_{a}bh$$
where *K* is the safety factor; *N* is the axial pressure, kN; *b* is the width of the section, m; *h* is the thickness of the section, m; *R_a_* is ultimate compressive strength of concrete or masonry, *R_a_* = 19 MPa; *φ* is the longitudinal bending coefficient of member, *φ* = 1; and *α* is the eccentric influence coefficient of the axial force, where, because the eccentricity is 0, *α =* 1.

#### 3.5.1. The Variation of Stress 

The void size was controlled to be 50 mm × 20 mm × 10 mm (length × width × depth), and the void behind was set at the vault, spandrel, hance and arch foot in turn. The stress values of lining the inner and outer walls at different positions were obtained, as shown in Figure 16. As can be seen from the figure, except for the tensile stress on the inside of the vault and the spandrel, other positions are subject to compressive stress, which is negative. From the numerical point of view, the outer stress is greater than the inner stress at the same monitoring point due to the concentration of stress in the void behind the lining. When the void was in the vault, the inner and outer stress values are the largest. 

#### 3.5.2. The Variation of Axial Force and Bending Moment 

The variation law of axial force and bending moment of the structure can be obtained by using Formulas (3) and (4), as shown in Figure 17. It can be seen from the figure that the axial force variation law of the lining structure was similar to the stress variation law of the outer lining void, and the maximum value was located at the arch waist. The bending moment in descending order of arch vault > spandrel > hance > arch foot, and the maximum bending moment of the lining structure was located at the vault.

#### 3.5.3. Variation of the Lining Safety Factor 

The lining safety factor can be obtained by using Formula (5). The variation law of the lining safety factor with void position is shown in the Figure 18, and its variation law is similar to the lining axial force. All safety factors are greater than two, meeting the requirements of safety standards, which are in descending order: arch foot > spandrel > vault > hance.

## 4. Conclusions

In this paper, a large-scale 1:10 indoor lining model was made by using a self-made horizontal test loading device to simulate the influence of the internal forces of the lining structure under different locations and different sizes of a void disease behind the lining. The main conclusions are as follows:(1)In terms of the stress law of the tunnel lining structure, the circumferential stress was generally greater than the axial stress at the same monitoring point, and the peak of the circumferential and axial tensile stresses appears at the vault, and the peak of the compressive stress appears at the arch waist. It can explain why the vault position was mainly damaged by stretching, and the arch position was mainly damaged by extrusion.(2)In terms of the law of the influence of the void position on the lining structure, when the void was in the vault, the stress change was more obvious, and when the void was in the position of the arch shoulder, the arch waist and the arch foot, compared with the lining without a hole, the ring and axial stress values of the same monitoring position of the lining do not change much. It was explained that when the void was in the vault, it had the greatest impact on the stress of the lining structure, and it had little effect on the stress of the lining structure when it was in other positions. The presence of void diseases can lead to varying degrees of increased stress values near the void location, and with the change of the position of the void, the stress value at the void was in order from large to small: the vault > the arch waist> the arch shoulder > the arch foot.(3)In terms of the law of the influence of the void depth on the lining structure, the circumferential stress and axial stress of the void and the inner wall of the vault increase with the increase of the void depth, and the increase of the vault void depth has a more obvious impact on the void and the stress of the inner wall of the vault.(4)In terms of the law of the influence of the length of the void on the lining structure, the circumferential and axial stresses of the inner wall of the lining vault increase with the increase of the void length, and the increase in axial stress increases. Stresses at other locations in the lining are not greatly affected by changes in void length.(5)The axial force variation law of the lining structure was similar to the stress variation law of the outer lining void, and the maximum value was located at the arch waist. The maximum bending moment of the lining structure was located at the vault and was in descending order of arch vault > spandrel > hance > arch foot.(6)The safety factor of the lining at different positions was greater than two, which meets the safety standard.

## Figures and Tables

**Figure 1 sensors-23-01403-f001:**
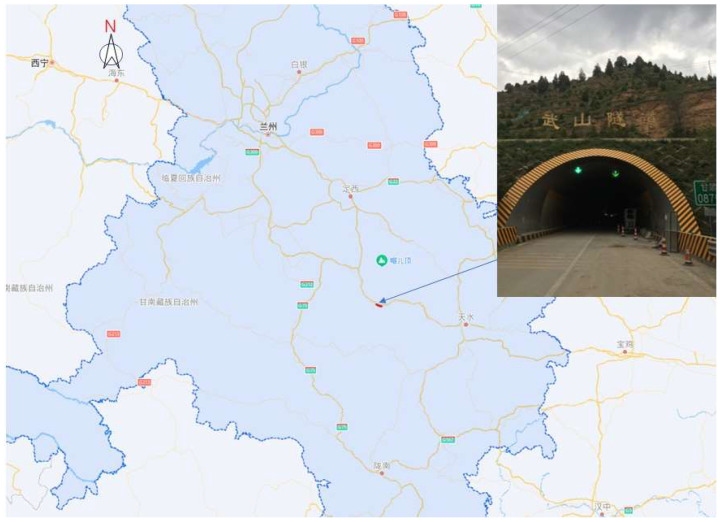
The Wushan Tunnel.

**Figure 2 sensors-23-01403-f002:**
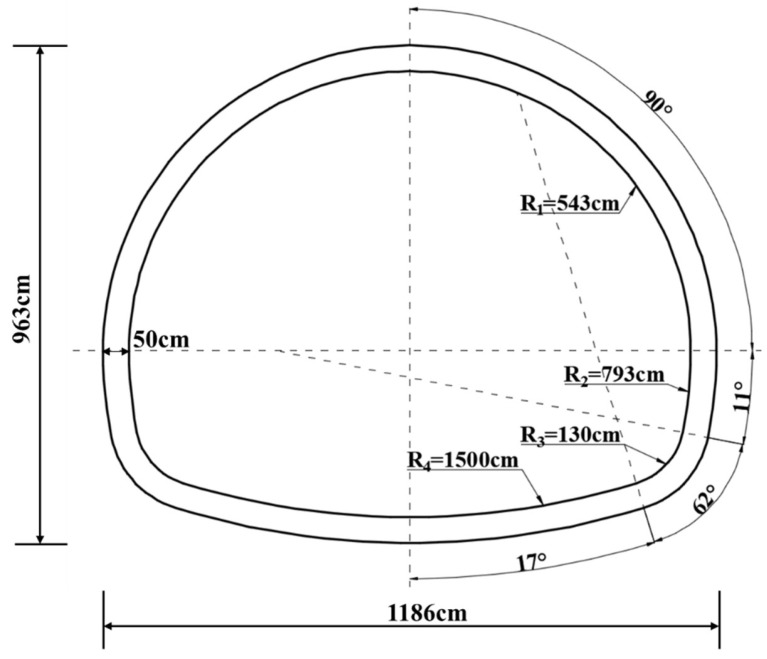
Tunnel secondary lining prototype section dimensions.

**Figure 3 sensors-23-01403-f003:**
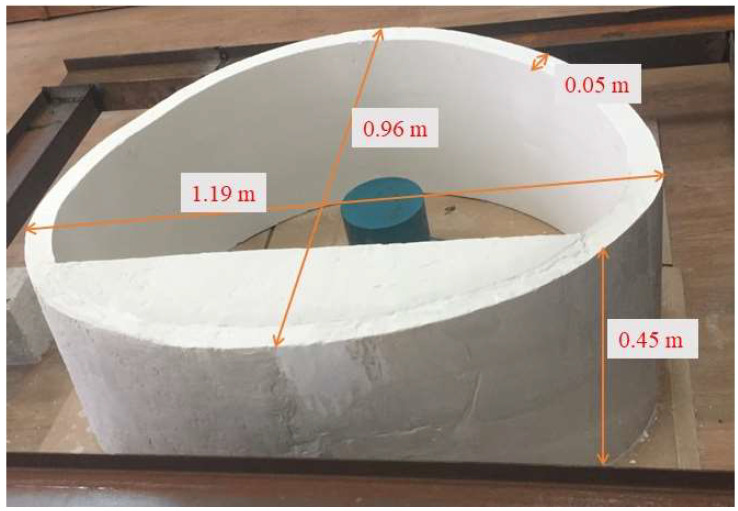
Similar lining model of tunnel.

**Figure 4 sensors-23-01403-f004:**
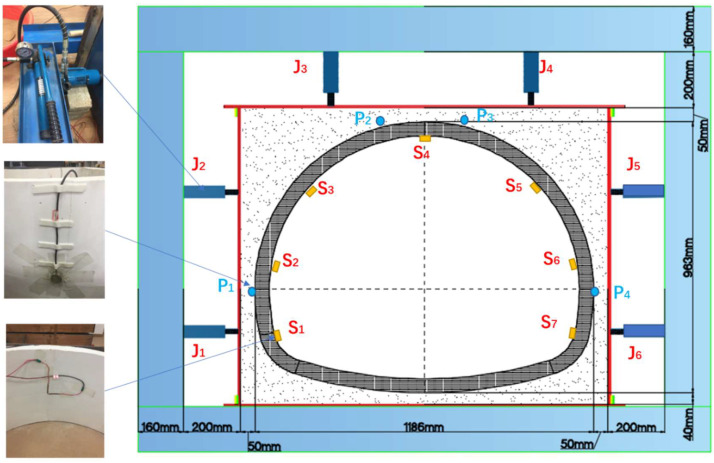
Schematic diagram of the test platform.

**Figure 5 sensors-23-01403-f005:**
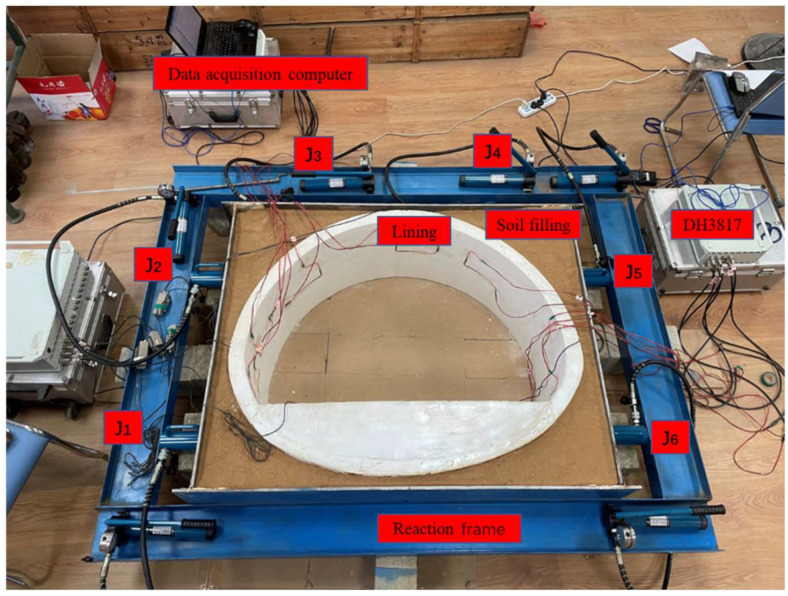
Self-made indoor similar model platform.

**Figure 6 sensors-23-01403-f006:**
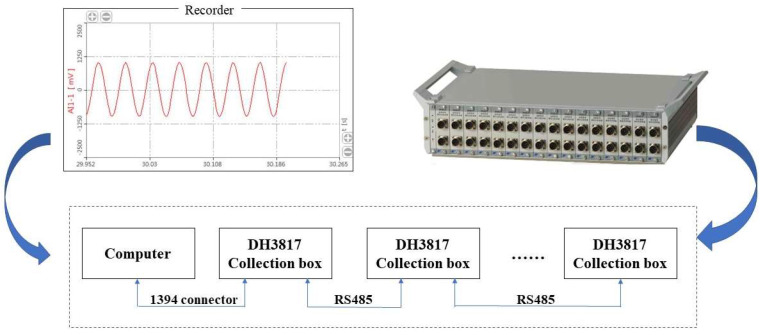
Working principle of DH3817.

**Figure 7 sensors-23-01403-f007:**
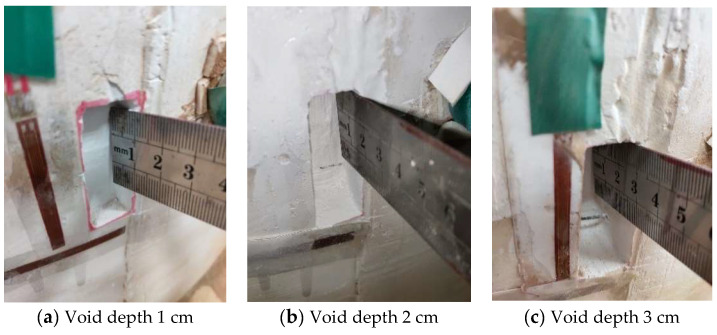
Vault void at different depths.

**Figure 8 sensors-23-01403-f008:**
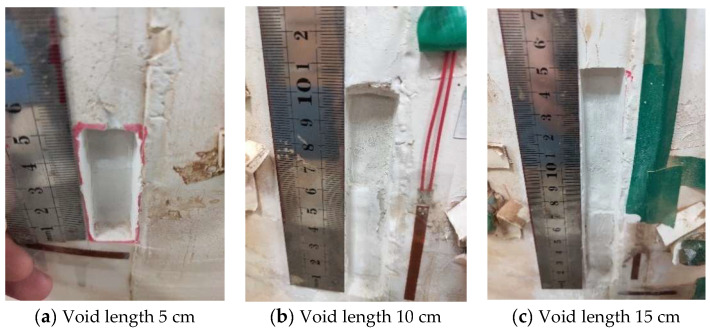
Vault void with different lengths.

**Figure 9 sensors-23-01403-f009:**
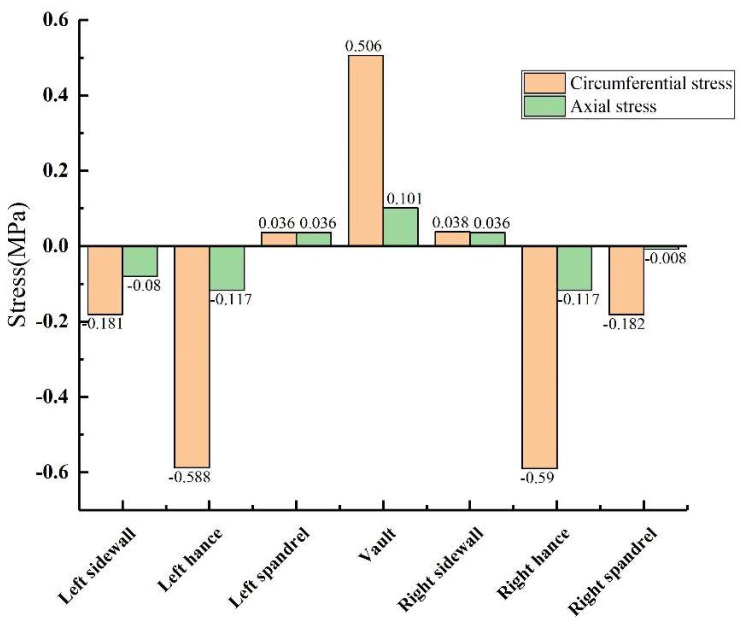
Lining without voids.

**Figure 10 sensors-23-01403-f010:**
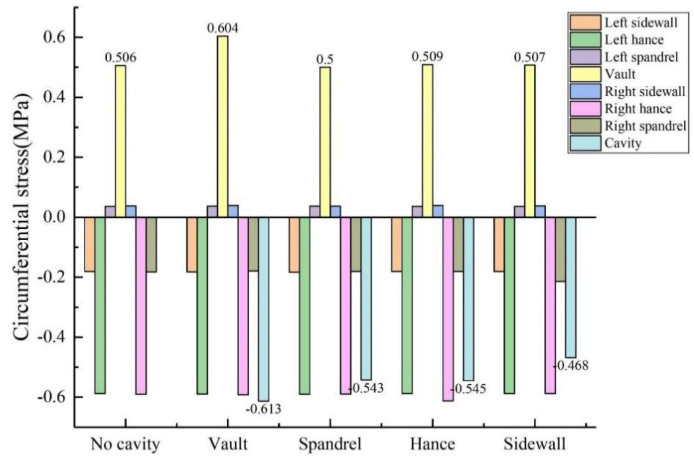
Variation of circumferential stress with void location.

**Figure 11 sensors-23-01403-f011:**
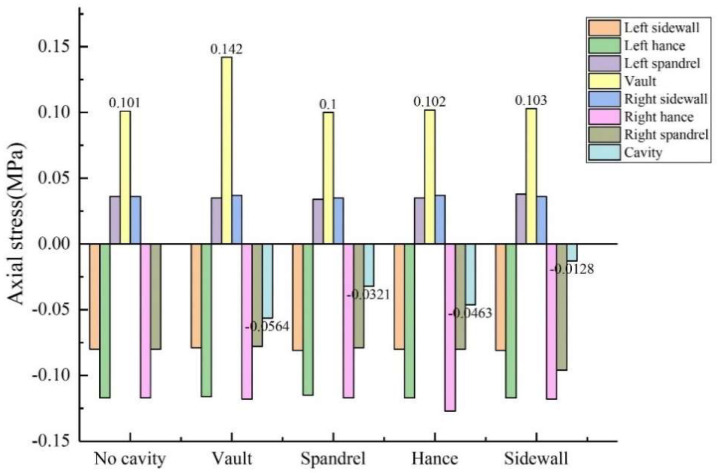
Variation of axial stress with void location.

**Figure 12 sensors-23-01403-f012:**
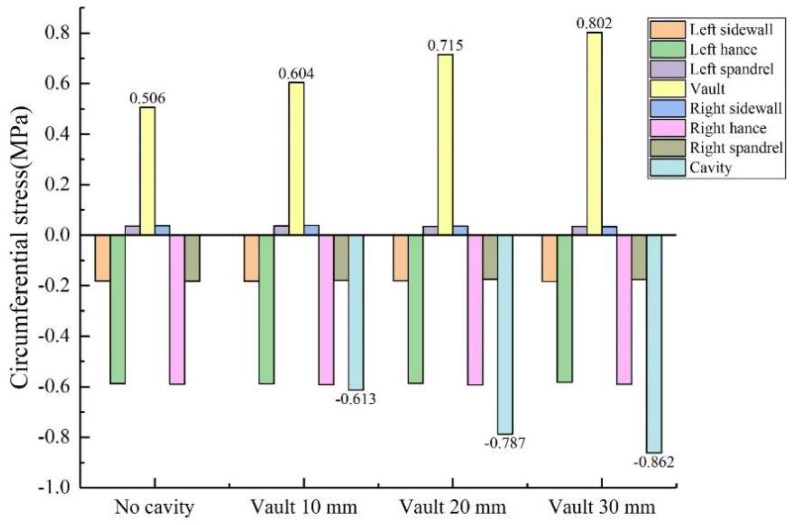
Variation of circumferential stress with vault void depth.

**Figure 13 sensors-23-01403-f013:**
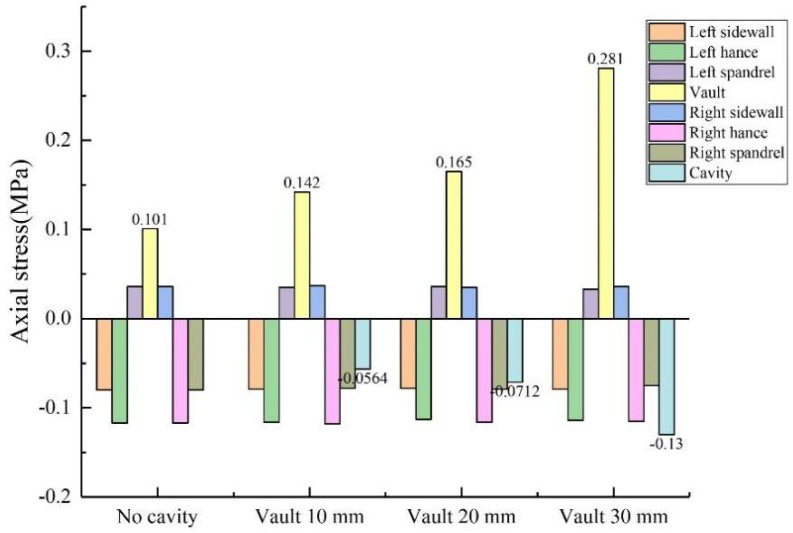
Variation of axial stress with vault void depth.

**Figure 14 sensors-23-01403-f014:**
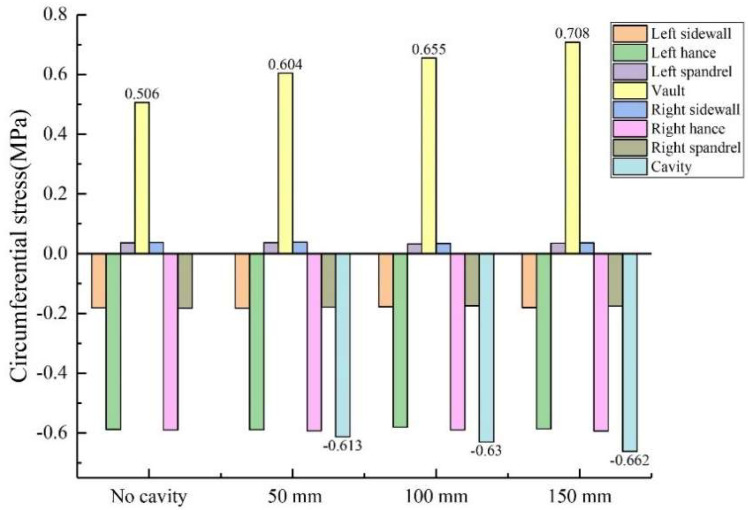
Variation of circumferential stress with vault void length.

**Figure 15 sensors-23-01403-f015:**
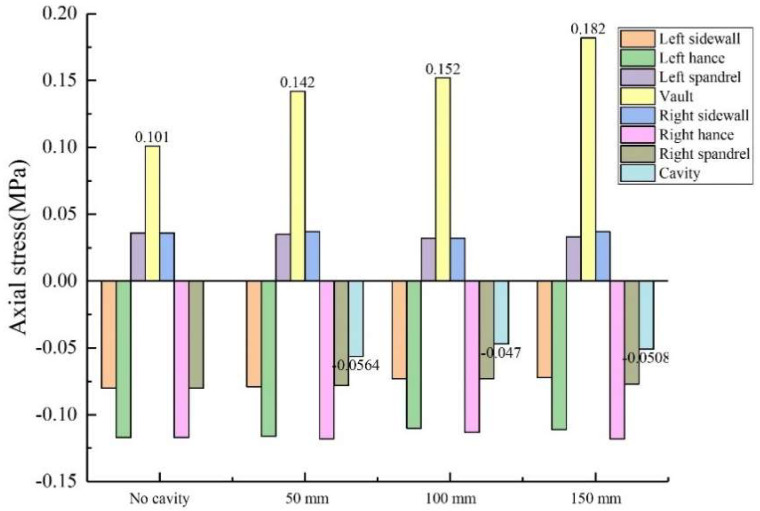
Variation of axial stress with vault void length.

**Figure 16 sensors-23-01403-f016:**
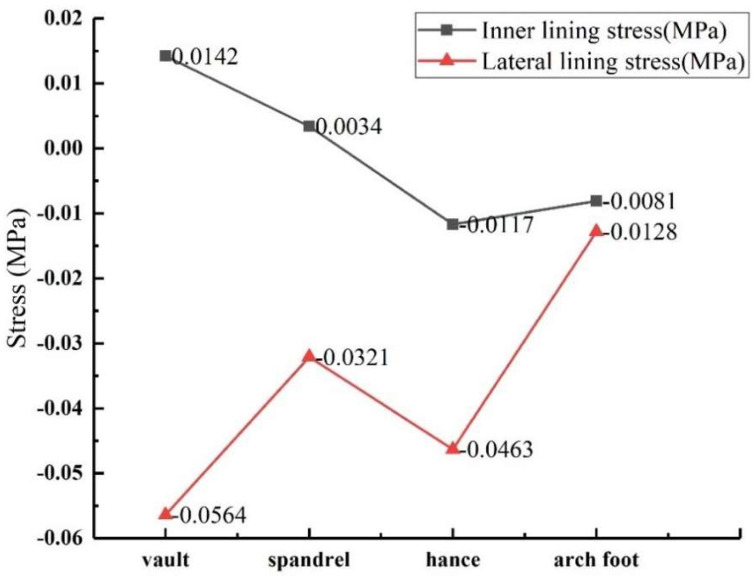
Lining stress variation law with void location.

**Figure 17 sensors-23-01403-f017:**
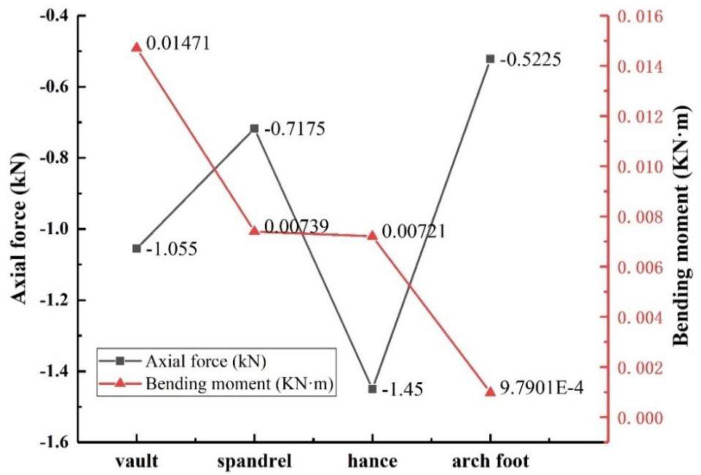
Variation law of axial force and bending moment with void location.

**Figure 18 sensors-23-01403-f018:**
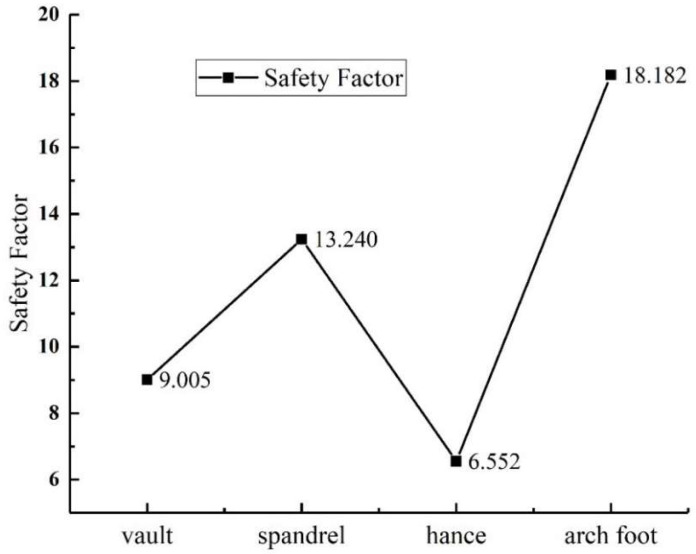
Variation law of safety factor of the lining structure with void location.

**Table 1 sensors-23-01403-t001:** Physical and mechanical indicators of lining.

Parameter	Severe *γ* (KN/m^3^)	Elastic Modulus *E* (GPa)	Poisson Ratio *μ*	Compressive Strength *R*_c_ (MPa)	Tensile Strength *R*_t_ (MPa)
Original material	25	28	0.2	16.7	1.78
Model material	11.8	2.651	0.2	1.674	0.168

**Table 2 sensors-23-01403-t002:** Value of surrounding rock pressure increase and decrease rate *i.*

Tunnel Width *B*(m)	*B <* 5	5 ≤ *B <* 14	14 ≤ *B <* 25	
Rate of pressure increase or decrease *i* in surrounding rock	0.2	0.1	Consider the excavation of the diversion hole during the construction process	0.07
			Up and down steps or one-time excavation	0.12

**Table 3 sensors-23-01403-t003:** Water level distribution pressure of deep buried tunnels.

Surrounding Rock Level	Ⅰ, Ⅱ	Ⅲ	Ⅳ	Ⅴ
Horizontal spread pressure (*e*)	0	<0.15*q*	(0.15~0.3)*q*	(0.3~0.5)*q*

**Table 4 sensors-23-01403-t004:** Test conditions for the void behind the lining.

Test Conditions	Void Location	Void Size (Length × Width × Depth)
1	There is no	/
2	vault	50 mm × 20 mm × 10 mm
3	vault	50 mm × 20 mm × 20 mm
4	vault	50 mm × 20 mm × 30 mm
5	vault	100 mm × 20 mm × 10 mm
6	vault	150 mm × 20 mm × 10 mm
7	spandrel	50 mm × 20 mm × 10 mm
8	hance	50 mm × 20 mm × 10 mm
9	The arch foot	50 mm × 20 mm × 10 mm

## Data Availability

The data that support the findings of this study are available from the corresponding author upon reasonable request.

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
