# Peer review of "Structural Stability Monitoring of Model Test on Highway Tunnel with Lining Backside Voids Using Dynamic and Static Strain Testing Sensors"

_sensors, 2023, doi:10.3390/s23031403_

Round 1

Reviewer 1 Report

In this study, a self-made indoor similar model loading apparatus is used to simulate a model test of the voids behind the lining of a highway tunnel for a typical tunnel project with damage. The impact of the lining cavity on the structural integrity of the highway tunnel is examined by varying the size and placement of the voids.

The manuscript fits well with the aim of Sensors journal and would be interesting for publication if the experimental methods were better described.

Specifically:

(1) The Authors should carefully review the grammar and syntax on the whole manuscript. There are instances where the text is not correct nor cohesive. For example, the Abstract starts with “The voids behind the lining will affect the interaction between surrounding rock and the lining, and even cause the structural instability, which will affect the tunnel safety”. There was no prior mention of a void, a lining, structural instability, or tunnel safety. Section 2.3 is very unclear. These kinds of problems are found throughout the whole manuscript and make it difficult to understand.

(2) In line 38, it would be interesting to specify which kind of numerical simulation was used. “Numerical simulation method” is too broad of a term.

(3) In lines 50-52, the Authors write “If only theoretical analysis and numerical simulation and other technical means are used for research, there are bound to be shortcomings”. It would be beneficial if the Authors could support their point of view with an example of such shortcomings.

(4)   Figure 2 could be made clearer. The important information regarding the arc are the values of their radii. Representing the radii by means of arrows does not add useful information – instead, it makes the Figure strange to look at. The magnitude of radius R_2, for example, is almost half of that of radius R_4, but it was represented with an arrow twice as big. It would be interesting if the Authors could remove the arrows and indicate the different radii by assigning codes to the different arc sections, for example.

(5)   In line 107 the Authors should include a reference to the similarity theory principle. Moreover, the Authors should develop further how the model values were deduced using this principle.

(6)  The Authors should develop in Table 1 on why the key parameters were chosen.

(7) In Figure 3, the Authors present variables T1 to T4 and Q1 to Q6, providing them only after Figure 4. The Authors could supply this variable close to the figure for the sake of clarity. "T" seems an odd choice of letter to represent a pressure sensor; the Authors could rename it to "P" for reasons of coherence. Moreover, in this Figure, there are dimensions without their magnitudes properly indicated, which makes the Figure hard to interpret.

(8) It is not clear if, according to the model similarity ratio, all the reduced model dimensions , or only the ones shown in Figure 4 were obtained. For example, in Figure 2, there are 7 dimensions represented, but in Figure 4, only 4 of them are mentioned. Have the other dimensions also been controlled? If yes, the Authors should specify it in the text. If not, shouldn’t these dimensions also be disregarded in Figure 2?

(9)  Through lines 186–192, wouldn't it be useful to delve into the details of this data acquisition system?

(10) Table 3 is divided by a page break. The Authors should adjust. Also, the column borders are not very clear.

(11) Figures 10 to 16 could be supplemented with a stress plot directly on the tunnel, to better visualize qualitatively the effect of each defect parameter.

(12) Since the objective of the Authors seems to obtain a qualitative grip of the effect of the different defects, it might be more interesting to plot percentile variation graphs in relation to the base case (No cavity).

(13) Figures 17 and 18 are hard to see. It would be better to change one of the data series line types and/or marker type and/or color.

(14) Figures 17 and 18 are related respectively to “Lining stress variation law with cavity location” and “Variation law of axial force and bending moment with cavity location”. However, in line 382, the “variation law of axial force and bending moment with cavity location” was introduced first. It would be interesting for the Authors to change the order of the figures.

(15) It is not clear what practical interest there is in calculating the variation law of axial force and bending moment.

According to what said above, the reviewer’s opinion is that the manuscript can be accepted for publication after the described major revisions.

Author Response

Thank you very much for your valuable advice. Please refer to the attachment for specific reply

Author Response

(The authors gave the same response as above.)

Reviewer 3 Report

The manuscript deals with an experimental investigation of Structural Stability Monitoring of Model Test on Highway Tunnel with Lining Backside Voids Using Dynamic and Static Strain Testing Sensors that can be improved further. The FEA analysis is recommended to perform and compared with the experimental data. A few more comments are embedded in the attached manuscript. The manuscript may not be recommended for the publication in its current form. 

Author Response

(The authors gave the same response as above.)

Round 2

Reviewer 1 Report

After having carefully read this last version of the manuscript, it can be said that the original paper has been correctly amended and improved by the Authors, providing adequate answers and addressing all the emerging issues.

The manuscript is interesting and fits well with the aim of the "Sensors" Journal. Therefore, the opinion of this reviewer is that it can be published in its current form.

Reviewer 3 Report

The manuscript may be recommended for publication.